# Convenient Agarose Preparation with Hydrogen Peroxide and Desulfation Process Analysis

**DOI:** 10.3390/md19060297

**Published:** 2021-05-23

**Authors:** Cong Zhang, Ding An, Qiong Xiao, Fu-Quan Chen, Yong-Hui Zhang, Hui-Fen Weng, An-Feng Xiao

**Affiliations:** 1College of Food and Biological Engineering, Jimei University, Xiamen 361021, China; zhangcong@jmu.edu.cn (C.Z.); anding@jmu.edu.cn (D.A.); xiaoqiong129@jmu.edu.cn (Q.X.); fqchenhy0109@jmu.edu.cn (F.-Q.C.); yhz@jmu.edu.cn (Y.-H.Z.); wenghuifen@jmu.edu.cn (H.-F.W.); 2National R&D Center for Red Alga Processing Technology, Xiamen 361021, China; 3Fujian Provincial Engineering Technology Research Center of Marine Functional Food, Xiamen 361021, China; 4Xiamen Key Laboratory of Marine Functional Food, Xiamen 361021, China

**Keywords:** agarose, hydrogen peroxide, desulfation analysis

## Abstract

Agarose is a natural seaweed polysaccharide and widely used in the medicine, food, and biological fields because of its high gel strength, non-toxicity, and electrical neutrality. The sulfate group is one of the main charged groups that affect the performance of agarose. In the present study, a simple, eco-friendly, and efficient method was explored for agarose preparation. After desulfation with hydrogen peroxide (H_2_O_2_), the sulfate content of agar reached 0.21%. Together with gel strength, electroendosmosis, gelling and melting temperature, the indicators of desulfated agar met the standards of commercially available agarose. Notably, the desulfated agar can be used as an agarose gel electrophoresis medium to separate DNA molecules, and the separation effect is as good as that of commercially available agarose. Further, the H_2_O_2_ desulfation process was analyzed. The addition of a hydroxyl radical (HO•) scavenger remarkably decreased the H_2_O_2_ desulfation rate, indicating that HO• has a certain role in agar desulfation. Sulfate content detection indicated that sulfur was removed from agar molecules in the form of sulfate ions (SO_4_^2−^) and metal sulfate. The band absence at 850 cm^−1^ indicated that the sulfate groups at C-4 of D-galactose in sulfated galactan were eliminated.

## 1. Introduction

Agarose is a deep-processed product of agar and widely used in the medicine, food, and biological fields [1,2]. It is a natural linear polysaccharide extracted from abundant red algae species, and is composed of 1,3-linked β-D-galactose and 1,4-linked 3,6-anhydro-α-L-galactose repeating units alternately [3]. This compound accounts for 70% of the agar composition with balance sulfated galactan [4]. Agarose contains less negatively charged groups than sulfated galactan, thus exhibiting good properties, such as low sulfate content, high gel strength, and low electroendosmosis (EEO). According to the agarose product overview provided by Sigma Company, agarose can be classified into three categories, such as low-, medium-, and high-EEO agarose. The low-EEO (0.05–0.13) agarose has a gel strength of at least 1200 g/cm^2^, the medium-EEO (0.16–0.19) agarose has at least 1000 g/cm^2^, and the high-EEO (0.23–0.26) agarose has at least 650 g/cm^2^. Moreover, the gelling and melting temperature ranges of Sigma agarose were 32–45 and 80–95 °C, respectively.

To obtain agarose, people usually use alkali treatment to extract agar with low sulfate content from red algae and then separate the agarose and sulfated galactan from agar [5,6]. Many agarose extraction methods are carried out by separating agarose and sulfated galactan. Some fractional precipitation methods such as the use of dimethyl sulfoxide, polyethylene glycol, ethylenediaminetetraacetic acid disodium salt (EDTA-Na_2_), and quaternary ammonium salts are utilized to separate agarose from agar [6,7,8,9,10]. However, the above methods produce a large volume of waste liquid, and the use of organic reagents will harm the environment. The ion exchange method is also used in the preparation of agarose [11,12]. In this method, the anion exchange resin can adsorb the negatively charged sulfated galactan components in agar to obtain agarose. The drawback of this method is that sulfated galactan is directly run off, causing a dramatic decrease in extraction yield. Besides, this method needs to dissolve agar in water and then remove water to obtain agarose. The time-consuming process increases the cost. Therefore, it can be considered by reducing the sulfate content of agar to improve the yield of agarose.

The content of sulfate groups is closely related to the gel strength, EEO and other basic properties of agarose. Generally, the lower the sulfate content is, the higher the gel strength and lower the EEO of agarose are [5,13]. Alkaline treatment is the traditional method of extracting agar, and this method removes the main sulfate contents of the galactopyranose unit by converting it to 3,6-anhydrogalactopyranose [5]. In addition, organic reagents such as chlorotrimethylsilane and pyridine are applied to remove the sulfate in sulfated galactose [14,15]. However, the above methods cannot meet the requirements of agarose because of the high sulfate content (above 1.0%) after treatment. In recent years, more environmentally friendly methods have emerged for the removal of sulfate from agar, such as the use of enzymes [16,17]. Enzymatic desulfation can significantly improve the quality of agar and reduce sulfate content to less than 0.2%. Moreover, no difference was observed between the DNA electrophoresis spectrum on the gel of the desulfated agar and commercial agarose. However, the source of enzymes greatly increases the cost of desulfation. Therefore, a simple, fast, efficient, low cost, and environmentally friendly method needs to be developed for the industrial production of agarose.

Hydrogen peroxide, an environmentally friendly reagent, is widely used in the fields of medicine, environmental treatment, and food [18,19,20]. Our research first found that hydrogen peroxide has a bleaching effect on agar, and we applied high-whiteness agar to jelly and medium colony count to explore the diversified applications of high-whiteness agar [21]. Moreover, hydrogen peroxide has a further desulfation effect on agar extracted by alkali treatment, and the quality of agar is notably improved [22,23]. Although hydrogen peroxide can effectively remove sulfate from agar, no systematic study has been conducted about the effects of different factors on the desulfation effect. Moreover, the evaluation of the potential application of desulfated agar as agarose and the analysis of hydrogen peroxide desulfation have not been clarified. These topics need to be studied for the preparation of agarose through further control, thus providing their research value and practical significance.

In our present work, desulfated agar was first obtained by exploring the desulfation effect of H_2_O_2_ under different reaction conditions (reaction pH, H_2_O_2_ concentration, ethanol concentration, reaction time, and reaction temperature). Then, to explore the agarose potential application performance of desulfated agar, we compared the basic properties and application performance of desulfated agar and commercial agarose. Next, H_2_O_2_ desulfation was explored by adding a free radical scavenger and analyzing the removal form of sulfur in agar molecules. Finally, FT-IR was performed to analyze the construction change of agar after H_2_O_2_ treatment.

## 2. Results and Discussion

### 2.1. Preparation of Desulfated Agar

H_2_O_2_ was easily decomposed into H_2_O and O_2_ in the reaction solution. This reaction is a balance of multiple chemical decompositions and reactions involving multiple ions and free radicals. The free radicals generated after the decomposition of hydrogen peroxide have strong oxidizing properties, which are widely studied in polysaccharide modification [24,25,26]. Moreover, hydrogen peroxide is often used as a desulfurizer in petroleum applications to prevent the sulfur dioxide produced by the subsequent combustion of petroleum from polluting the atmosphere [27,28]. Based on our previous research, HOO- decomposed by hydrogen peroxide is a very strong nucleophile, which can react with pigments for the purpose of bleaching agar [21]. In addition, hydrogen peroxide can effectively remove sulfate from agar at appropriate reaction conditions [22,23]. However, no systematic study has been conducted to determine the effects of different factors on the desulfation effect. Thus, to systematically explore the effect of hydrogen peroxide on agar desulfation under different conditions, we used sulfate content and gel strength as indicators and investigated different factors such as reaction pH, H_2_O_2_ concentration (%), ethanol concentration (%), reaction time (h), and reaction temperature (°C). Multiple ions and free radicals were generated when H_2_O_2_ was present in the reaction solution, possibly causing the difference in sulfate content [29]. Figure 1a shows that the minimal sulfate content of desulfated agar was achieved when the reaction pH reached 9.0. In this pH reaction condition, HO• is the main active component of desulfation [30]. With the increased pH from 9.0 to 11.0, the sulfate content increased sharply from 0.28% to 0.57%. The reason of this phenomenon might be attributed to producing HOO- to achieve the purpose of bleaching agar [21]. In addition, to make the agar powder more uniformly dispersed in the reaction solution, we added ethanol to the reaction system. Figure 1c shows the effect of different ethanol concentrations on hydrogen peroxide desulfation. With the increase in ethanol concentration, the sulfate content of desulfated agar gradually increased. This phenomenon might be related to the quenching effect of ethanol in the reaction system on hydroxyl radicals [31]. Finally, desulfated agar was obtained at the reaction pH of 9.0, H_2_O_2_ concentration of 2%, ethanol concentration of 30%, reaction time of 2 h, and reaction temperature of 40 °C. Under the above reaction conditions, the sulfate content of desulfated agar obtained reached 0.21%, and the desulfation rate reached 76.7%.

The desulfation rate with different methods is summarized in Table 1. Alkali treatment is a traditional method for extracting agar, and this method can remove the sulfate by converting L-galactose-6-sulfate in the agar structure to the 3,6-anhydro-L-galactose form. The desulfation rates obtained through this method exceeded 35%, as shown in Table 1. Polysaccharides extracted from seaweed can also achieve the desulfation purpose after treatment with organic reagents, through which the desulfation rate can exceed 60% [15]. Although alkali treatment and organic reagent treatment can achieve the purpose of desulfation, the sulfate content of the final samples obtained by these two methods was much higher than the standard of agarose. Enzymatic treatment is a desulfation method that has emerged in recent years [17]. Table 1 shows that the sulfate content of this method can be reduced to less than 0.20%, and the desulfation rate can exceed 85%. However, the disadvantage of this method is that the source of the enzyme increases the production cost. Therefore, a method with high desulfation efficiency and low cost should be determined. By using H_2_O_2_ treatment in our study, the sulfate content of desulfated agar reached 0.21%, and the desulfation rate was 76.7%.

After desulfation with H_2_O_2_, the sulfate content of desulfated agar met the sulfate content standard required by commercially available agarose. At present, there are many methods for extracting agarose, and the traditional method of extracting agarose is based on the principle of separating agarose from agar. Firstly, DEAE-cellulose exchange resin adsorbs acidic polysaccharides to achieve the purpose of separating and purifying agarose [13]. The production process of this method is not only complicated, but also the use of anion exchange resin greatly increases the cost. Secondly, polyethylene glycol (PEG) is also used for extracting agarose by dissolving sulfated galactan in the polyethylene glycol (PEG) solution [7]. It will take a long process time if the high-quality agarose is extracted by this way because the process needs to be repeated many times. Next, ethylene diamine tetraacetic acid (EDTA) is combined with anion exchange resin to extract agarose [6]. The combination of these two methods will result in a significant drop in the yield of agarose.

In our study, the method of using hydrogen peroxide to extract agarose is simple, fast, efficient, low-cost, and environmentally friendly. Finally, as shown in Table 2, it was found that the process of preparing agarose by hydrogen peroxide has the highest yield.

Therefore, the basic properties and potential application of desulfated agar were further studied and compared with commercial agarose products in the next experiments.

### 2.2. Analysis of the Basic Properties of Desulfated Agar and Commercial Agarose Products

The basic indicators for evaluating agarose are sulfate content, gel strength, EEO, gelling temperature, and melting temperature, and these indicators determine the quality of agarose [8,10,33]. The sulfate content, gel strength, and EEO of raw agar, desulfated agar, and commercial agarose are shown in Table 3. Sulfate is the main charged group in the monosaccharide units of natural agarose, and sulfate can be used as a purity indicator of agarose [34]. As shown in Table 3, the sulfate content of raw agar was higher than that of desulfated agar, and this value was reduced from 0.90% of raw agar to 0.21%. For desulfated agar, the sulfate content did not differ from that of agarose (Sigma A9793, 0.20%) and was close to that of agarose (Quanshijin GS201-01, 0.18%).

Gel strength was negatively correlated with sulfate content because sulfate can increase the mutual repulsion between agar molecules and hinder the aggregation of double helices. During gel formation, the three-dimensional network structure is relatively loose, thus decreasing the gel strength [5]. Table 3 shows the gel strength of raw agar, desulfated agar, and commercial agarose. In comparison with raw agar (903 g/cm^2^), the gel strength of desulfated agar (1023 g/cm^2^) increased by 13.3%, but a certain gap remained compared with commercial agarose. 

EEO is one of the quality standards of agarose, and it refers to the movement of liquid in gel [2,33]. The anionic group in agarose gel is fixed on the matrix and cannot move, but it can prevent the decomposable counter cation from migrating to the cathode in the matrix. Results show that the EEO of desulfated agar (0.238) was 48.8% lower than that of raw agar (0.465), and desulfated agar met the standard of Sigma’s high-EEO agarose (EEO: 0.23–0.26).

Figure 2 shows the changes in G’ and G” of the four samples as a function of temperature. The gelling temperature (Tg) of the sample was obtained at a point in which G’ and G” intersected during cooling ramp (Figure 2a), and melting temperature (Tm) was in a point during heating ramp (Figure 2b) [35]. The Tg of desulfated agar was observed at 40.4 °C, and this value was lower than that of raw agar (41.9 °C). During heating ramp, the melting temperature of desulfated agar was determined at (91.0 °C), and this value increased slightly compared with that of raw agar (90.5 °C). The increase in Tm of desulfated agar could be ascribed to the decrease in sulfate content that enhanced the interaction of hydrogen bonds and agar molecules, which required more energy to melt it.

### 2.3. Application Performance Analysis

Agarose gel is a commonly used electrophoresis support medium because of its high gel strength, non-toxicity, and ability to maintain a relatively pure background after rapid staining and decolorization [36]. The most important factor that affects the results of DNA agarose gel electrophoresis is EEO. The higher the EEO is, the more negatively charged substituents on the agarose chain tend to move to the anode during electrophoresis, thus hindering the movement of the sample to the cathode, resulting in failure to form clear electrophoretic bands [1].

The application performance of raw agar, desulfated agar, and commercial agarose in the electrophoresis process was studied through agarose gel electrophoresis, and the results are shown in Figure 3. Raw agar (Figure 3a) was used as the electrophoresis medium to separate the DNA fragments of different molecular weights. When the molecular weight of the DNA marker exceeded 1000, the electrophoresis bands began to blur. By contrast, when desulfated agar (Figure 3b) was used as the electrophoresis medium, the DNA markers of different molecular weights could be separated well. In addition, as can be seen from the agarose gel electrophoresis diagram (Figure 3b–d), the separation effect of the desulfated agar is as good as that of commercially available agarose and it can be used as the agarose gel electrophoresis medium to separate DNA molecules. Sulfate is one of the main negatively charged groups in the agar molecule [37]. The reduction in the sulfate content of desulfated agar decreased electro-infiltration, and the electrophoresis band became clearer.

### 2.4. Desulfation Process Analysis

Considering the role of hydroxyl radicals (HO•) in polysaccharide oxidation and oil oxidative desulfurization, the ratio of HO• in the reaction system changed with the change in reaction conditions, and this condition may cause a difference in sulfate content [24,38]. Therefore, to reveal the roles of H_2_O_2_ in the desulfation process, we conducted a series of experiments for comparison. As shown in Figure 4a, when only hydrogen peroxide was present in the reaction system, the sulfate content of the desulfated agar (0.21%) was reduced by 76.7% relative to the sulfate content of raw agar (0.9%). Furthermore, HO• trapping experiments were conducted in the presence of H_2_O_2_ by adding an excess of ascorbic acid (Vc), isopropanol (IPA), and dimethyl sulfoxide (DMSO). The hydrogen peroxide desulfated rate decreased from 76.7% to 39.5%, 41.6% and 22.8%, and HO• had a certain role in desulfation [39,40,41].

Moreover, to explore the whereabouts of sulfur in the agar molecule, we detected the sulfate content in the reaction solution and washing solutions. Figure 4b shows that the amount of sulfate radicals removed by the desulfated agar was far greater than the sum of the sulfate radicals in the reaction solution and the washing solution. Part of the sulfur in the agar molecule was connected to the polysaccharide backbone in the form of sulfurous acid, and the other part was combined with the metal ion in the form of sulfate (Figure 5) [42]. Therefore, the sulfur from raw agar was removed in the form of ions (SO_4_^2−^), and most compounds were in the form of metal ion sulfate (e.g., CaSO_4_, MgSO_4_ et al.).

Figure 4c shows the spectra of the raw and desulfated agar in the wavelength range of 4000 cm^−1^ to 400 cm^−1^. Data revealed the characteristic bands of agar-type polysaccharides (1250, 1072, 930, and 893 cm^−1^). The absorption peaks of raw agar and desulfated agar at 3391, 2932, and 1640 cm^−1^ were attributed to the stretching vibrations produced by O–H, C–H, and C–O, respectively [35]. The regions at approximately 1250 cm^−1^ were attributed to the asymmetric stretching of the sulfate ester group. In comparison with raw agar, the desulfated agar had a smaller peak, indicating its lower sulfate content. The regions at approximately 1070 and 890 cm^−1^ were equivalent to the skeleton of galactan and agar specific band. The band at 930 cm^−1^ was assigned to the vibration of the C–O–C bridge in 3,6-AG. Considering that the infrared spectrum detection line cannot detect sulfate at different positions in raw agar, to increase the sulfate content, we used Wang’s method to extract the sulfated galactan with a higher sulfate content from the raw agar. The spectra of the samples before and after the modification are shown in Figure 4d. The position of the sulfate group of agar-type polysaccharide was identified by the bands at 800–850 cm^−1^ [17]. The absence of the band at 850 cm^−1^ of modified sulfated galactan indicates that the sulfate groups of C-4 of D-galactose were eliminated.

## 3. Materials and Methods

### 3.1. Materials

Agar powder was provided by Green Fresh Food Stuff Co., Ltd (Zhangzhou, China). Hydrogen peroxide (30 wt%) was obtained from Xilong Science and Technology Co., Ltd. (Shanghai, China). Agarose (Sigma: A9793; Quanshijin: GS201-01) was obtained separately from Sigma (Saint Louis, MO, USA) and Quanshijin Co., Ltd. (Beijing, China).All reagents used in the experiments were obtained from Xilong Science and Technology Co., Ltd. (Shanghai, China).

### 3.2. Preparation of Desulfated Agar with Hydrogen Peroxide

Agar powder (25 g) and a certain amount of hydrogen peroxide (30 wt%, 0–6%) were added to 250 mL of a certain concentration of ethanol solution (30–90%) at different temperatures (30–70 °C) for 1–3 h. The solution pH was adjusted to 3.0–11.0 during the reaction. Then, the agar powder was washed several times with distilled water to remove residual hydrogen peroxide. Next, the sample was dried at 55 °C. Finally, the process condition for desulfated agar was obtained.

### 3.3. Preparation of Sulfated Galactan from Agar

Sulfated galactan was extracted as described by Wang’s method [17]. Agar powder (10 g) was dissolved in 500 mL of 6.5 g/L EDTA-Na_2_ solution and stirred for 4 h at 60 °C. After the reaction, the supernatant was collected by filtration and then concentrated using rotary evaporator. The sulfated galactan was precipitated with ethanol (1:3, *v/v*). Then, sulfated galactan was redissolved in deionized water and dialyzed for 48 h by using a dialysis bag. Finally, sulfated galactan from agar powder was isolated repeatedly as described above.

### 3.4. Desulfation of Sulfated Galactan

Sulfated galactan (10 g) and 5 mL of hydrogen peroxide (30 wt%) were added to 100 mL of 45% ethanol solution at 40 °C for 2 h. The solution was adjusted to pH 9.0 during the reaction. Then, sulfated galactan was washed several times with distilled water to remove residual hydrogen peroxide. Finally, the sample was dried at 55 °C.

### 3.5. Sulfate Content

The sulfate content was determined by using the BaCl_2_ turbidimetric method with slight modification [43]. Exactly 300 mg of the sample was hydrolyzed in 25 mL of 1 M HCl at 100 °C for 5 h. After cooling to room temperature, the sample was treated with activated carbon for decolorization and then filtered to obtain digestive juice. Exactly 1 mL of sample digestive juice was transferred to a tube containing 3 mL of BaCl_2_–gelatin reagent (0.5% gelatin solution containing 1% BaCl_2_). The absorbance at 360 nm was measured after blending for 10 min. Blank sample was prepared using the above steps except that distilled water was used instead of agar sample. The standard curve was prepared using K_2_SO_4_ at the concentration range of 0–1 mg/mL (0–0.12 mg/mL SO_4_^2−^). The sulfate content was calculated by substituting the measured absorbance value into the standard curve.

### 3.6. Gel Strength

The gel strength was measured as described by Lee, Namasivayam and Ho [44]. A 1.5% (*w/v*) agar solution was poured into a cylindrical mold with 3 cm diameter and 2.5 cm height and then held at 20 °C for 12 h. Then, a texture analyzer (Stable Micro System, Surrey, UK) with a load cell of 5 kg, cross-head speed of 1 mm/s, and equipped with a 1.27 cm-diameter flat-faced cylindrical Teflon^®^ plunger (Stable Micro System, Surrey, UK) was used. The maximum force (gram) was recorded when the plunger had penetrated 5 mm into the agar gels. Gel strength was calculated and expressed in g/cm^2^.

### 3.7. EEO

EEO was described by Hu with a slight modification [13]. First, 150 mg of the sample was dissolved in barbitone buffer solution (pH = 8.6) to prepare an electrophoresis gel plate. Then, gel loading solutions were prepared by dissolving 50 mg bromothymol blue in 8 mL of barbitone buffer solution (pH = 8.6), and the sample was filtered. Dextran T70 (100 mg) and human albumin (100 mg) were added into the filtrated solution, and 10 mL of solution was prepared by dissolving in barbitone buffer solution (pH = 8.6). Next, 6 μL of the sample configuration was added into the electrophoresis gel plate hole, and the voltage was kept at 85 V for 1 h. Finally, the gel plate was washed off with decolorizing reagent (5% CH_3_COOH and 95% ethanol were mixed equally) for 15 min, dipped in color-developing reagent (100 mg amido black was added into 1000 mL of 95% ethanol) for 30 min, and washed off with decolorizing reagent again. Then, the EEO was expressed as follows:EEO = OD/ (OD + OA)(1)
where OD is the distance from the blue spot on the cathode to the sample hole, and OA is the distance from the white spot on the anode to the sample hole.

Gel electrophoresis was performed using raw agar, desulfated agar, and commercial agarose. The experiment was performed with 1.0% gel and DNA marker 1000, 2000, 5000 and 10,000 by using mini-sub cell GT 8 gel rigs [45].

### 3.8. Gelling Temperature (Tg) and Melting Temperature (Tm)

The gelling and melting temperatures of the samples were determined using a DHR-2 rheometer (TA Instruments, Newcastle, DE, USA) [35]. Exactly 3 mL of 1.5% concentration of hot agarose solution samples was prepared for each analysis. The agarose samples were assessed on a stress-controlled rheometer (ARG2, TA Instruments, Newcastle, DE, USA) equipped with a serrated parallel-plate (Reologica Instruments AB) with a diameter of 60 mm and a gap of 1.0 mm. The storage modulus (G’) and loss modulus (G”) of the gels were obtained as a function of temperature by setting the temperature programs used at a cooling (80–30 °C) and heating (30–95 °C) rate of 2 °C/min.

### 3.9. Spectroscopy Methods

For obtaining the FT-IR spectra, the samples were prepared by mixing the sample and KBr tableting at a ratio of 1:200 in a dry environment. Then, the FT-IR spectra were obtained at the wavenumber range of 4000–500 cm^−1^ with a Nicolet Is50 infrared spectrometer (Thermo Fisher Scientific, Wal than, MA, USA).

### 3.10. Statistical Analysis

All determinations were made in triplicate by using three parallel samples. Statistical analysis was performed using the Statistical Package for Social Sciences (SPSS 17.0 for Windows, SPSS Inc., Chicago, IL, USA) software. Data were analyzed by analysis of variance and the differences between means by the Duncan’s multiple range tests.

## 4. Conclusions

The method of convenient preparation agarose was successfully explored. Agar was modified with H_2_O_2_ and, after optimization, the sulfate content of desulfated agar reached 0.21%. The basic index sulfate content, gel strength, EEO, gelling temperature, and melting temperature of desulfated agar met the standards of commercially available agarose. Further, the performance of desulfated agar electrophoresis gel was studied. The DNA markers of different molecular weights can be separated better than that of raw agar. The HO• scavenging experiment showed that HO• has a certain role in the desulfation. The sulfate content detection results show that sulfur in raw agar was removed in the form of ions (SO_4_^2−^) and metal sulfate. FT-IR results show that the absence of the band at 850 cm^−1^ of modified sulfated galactan indicates that the sulfate groups of C-4 of d-galactose were eliminated. Therefore, the method of preparing agarose with H_2_O_2_ could be effectively exploited, and the desulfation process analysis above could provide a theoretical basis for further studies.

## Figures and Tables

**Figure 1 marinedrugs-19-00297-f001:**
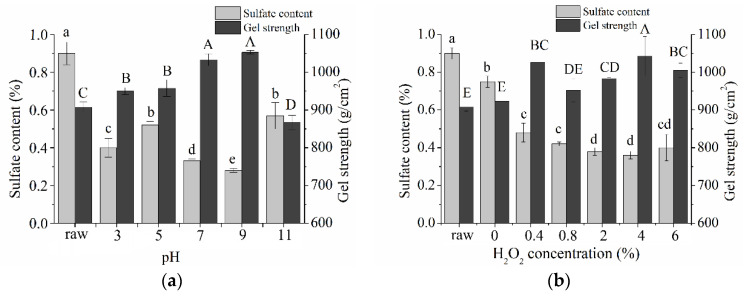
Effects of reaction conditions on agar desulfation. (**a**) Reaction pH; H2O2 concentration 4%; Ethanol concentration 30%; Reaction time 2 h; Reaction temperature 30 °C. (**b**) H2O2 concentration; pH 9.0; Ethanol concentration 30%; Reaction time 2 h; Reaction temperature 30 °C. (**c**) Ethanol concentration; pH 9.0; H2O2 concentration 2%; Reaction time 2 h; Reaction temperature 30 °C. (**d**) Reaction time; pH 9.0; H2O2 concentration 2%; Ethanol concentration 45%; Reaction temperature 30 °C. (**e**) Reaction temperature; pH 9.0; H2O2 concentration 2%; Ethanol concentration 45%. The reaction was carried out by changing one parameter in each case while holding the four other parameters constant (pooled data from three experiments are presented as means ± standard deviation of the mean (*n* = 3)). “raw” indicates untreated raw agar. a, b, c, d, e on the bar graph indicate significant differences in sulfate content, and A, B, C, D indicate significant differences in gel strength (*p* < 0.05).

**Figure 2 marinedrugs-19-00297-f002:**
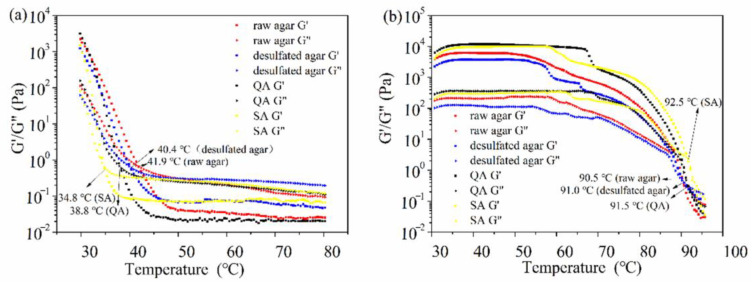
Temperature dependence of storage modulus G’ and loss modulus G” of the raw agar, desulfated agar, and commercial agarose (SA: Sigma agarose A9793; QA: Beijing Quanshijin agarose GS201-01) during (**a**) cooling ramp and (**b**) heating ramp. The point of the arrow indicates the gelation temperature (**a**) and melting temperature (**b**) of the samples.

**Figure 3 marinedrugs-19-00297-f003:**
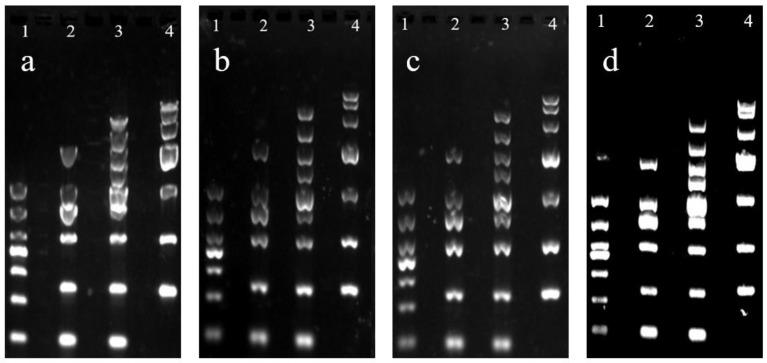
Agarose gel electrophoresis by using the gel of (**a**) raw agar, (**b**) desulfated agar, (**c**) commercial agarose (Beijing Quanshijin Biotechnology Co., Ltd., GS201-01), and (**d**) commercial agarose (Sigma A9793). (Lanes 1, 2, 3, and 4 represent molecular weights of 1000, 2000, 5000, and 10,000 DNA marker, respectively).

**Figure 4 marinedrugs-19-00297-f004:**
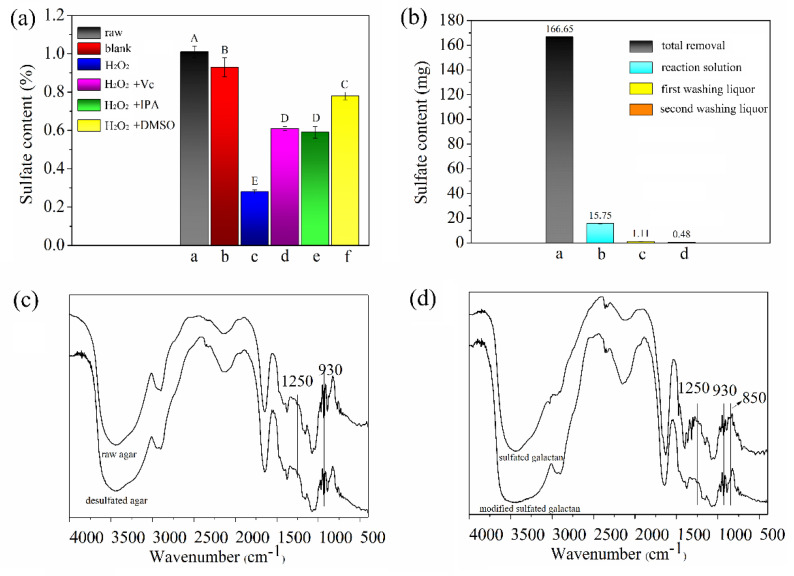
(**a**) Sulfate content of different reaction conditions (a: untreated raw agar, b: blank without H_2_O_2_, c: only H_2_O_2_, d: H_2_O_2_ and Vc, e: H_2_O_2_ and IPA, f: H_2_O_2_ and DMSO). (**b**) Sulfate content of total removal, reaction solution, first washing liquor and second washing liquor. (**c**) FT-IR of the raw agar and desulfated agar. (**d**) FT-IR of the sulfated galactan and modified sulfated galactan with hydrogen peroxide. A, B, C, D, E indicate significant differences in sulfate content (*p* < 0.05).

**Figure 5 marinedrugs-19-00297-f005:**
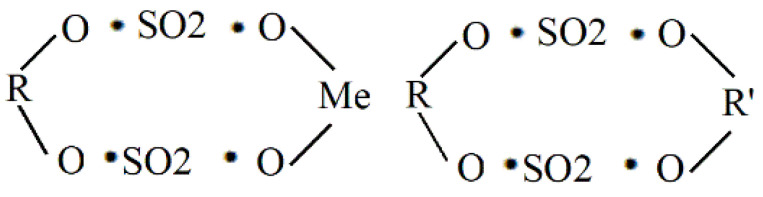
Main form of sulfur in agar molecules (R/R’: polysaccharide backbone, Me: Mg, Ca et al.).

**Table 1 marinedrugs-19-00297-t001:** Influence of different treatment methods on sulfate radical content.

Method	Source of Red Algae	Origin	Sulfate Content (%)		Desulfation Rate (%)	Reference
			**Before**	**After**		
Alkaline treatment	*Gracilariopsis. chorda*	Chilean	2.06	0.95	53.88	[32]
	*Gracilariopsis. chorda*	Argentinian	2.22	1.43	35.59	[32]
Organic reagent treatment	*Gigartina decipiens*		11.6	4.6	60.34	[15]
Enzymatic treatment	*Gelidium. amansii*	24.8 N, 118.6 E	1.11	0.16	85.59	[17]
	*Gracilaria. lemaneiformis*	35.7 N, 120 E	1.52	0.17	88.82	[17]
H_2_O_2_ treatment	Commercial agar	Greenfresh (Fujian) Food Stuff Co., Ltd.	0.71	0.42	40.85	[23]
			0.79	0.21	73.42	[22]
			0.90	0.21	76.67	Current study

**Table 2 marinedrugs-19-00297-t002:** Yield of different agarose extraction methods.

Method	Yield (%)	Reference
DEAE-cellulose	32.5 56.5	[13]
Polyethylene glycol (PEG)	30–45	[7]
EDTA-anion exchange resin	11.3	[6]
H_2_O_2_	74.0	Current study

**Table 3 marinedrugs-19-00297-t003:** Physiochemical properties of raw agar, desulfated agar, and commercial agarose.

	Raw Agar	Desulfated Agar	Agarose ^1^	Agarose ^2^
Sulfate content (%)	0.90 ± 0.01 ^a^	0.21 ± 0.01 ^b^	0.18 ± 0.01 ^bc^	0.20 ± 0.02 ^b^
Gel strength (g/cm^2^)	903 ± 8 ^C^	1023 ± 45 ^B^	1323 ± 11 ^A^	1340 ± 14 ^A^
EEO	0.465 ± 0.010 ^α^	0.238 ± 0.013 ^β^	0.181 ± 0.006 ^γ^	0.233 ± 0.013 ^β^

Physiochemical properties of raw agar, desulfated agar, and commercial agarose; Agarose ^1^: Obtained from Beijing Quanshijin Biotechnology Co., Ltd., GS201-01. Agarose ^2^: Obtained from Sigma A9793. Mean ± S.D. (n = 3). Superscripts a, b, c indicate significant differences in sulfate content. A, B, C indicate significant differences in gel strength and α, β, γ indicate significant differences in EEO (*p* < 0.05).

## Data Availability

Data are contained within the article.

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
