# Peer review of "Convenient Agarose Preparation with Hydrogen Peroxide and Desulfation Process Analysis"

_marinedrugs, 2021, doi:10.3390/md19060297_

Round 1

Reviewer 1 Report

Dear Authors,

In my opinion, studies presented by Cong Zhang and co-authors are quite interesting. Unfortunately, I think that manuscript is prepared in a careless manner and chaotic. The manuscript requires a lot of corrections. In details:

  1. Reading this manuscript it is difficult to deduce why the proposed method is good or better than others. For example, sulfate content after desulfation is 0.21 % for agar described in lit 22 as well as in this work!
  2. The authors repeat the same sentences, e.g. Abstract lines 14-16 and Introduction lines 31-33 or lines 74-75 and lines 96-97. The entire manuscript should be reviewed.
  3. The titles of Tables 1 and 2 need to be corrected.
  4. Why is part of the second row in Table 1 in bold?
  5. Figure 1: the lack of explanation for the signs: a, b, .... A, B, etc.
  6. Line 174-175: the lack of explanation of Tg and Tm.
  7. In Table 2, the letters a, b, c are in superscript; but nowhere is there an explanation of what they mean.
  8. There are graphs on page 4 (Figure 1) and their description is on page 7. The reader would expect a description next to the graphs, it will certainly be more convenient.
  9. What the bar marked as raw is about in the graphs (Figure 1) ? It should be explain.

Best wishes,

Reviewer

Author Response

Dear Reviewer:

Thanks a lot for your suggestions to improve the quality of our manuscript. Please see the attachment.

Best wishes.

Yours sincerely.

An-Feng Xiao

E-mail address: xxaaffeng@jmu.edu.cn

Reviewer 2 Report

The manuscript submitted by Cong Zhang et. al. investigates the novel method of obtaining desulfated agar, using hydrogen peroxide. In this work, desulfated agar was first obtained by exploring the desulfation effect of H2O2 under different reaction conditions (reaction pH, H2O2 concentration, ethanol concentration, reaction time, and reaction temperature). Then, to explore the agarose potential application performance of desulfated agar, authors compared the basic properties and application performance of desulfated agar and commercial agarose. Next, H2O2 desulfation process was explored by adding free radical scavenger and analyzing the removal form of sulfur in agar molecules. Finally, FT-IR was performed to analyze the construction change of agar after H2O2 treatment. The objectives was stated clearly. This research has provided a lot of information that may allow this novel method to be used in desulfation process under ecological conditions. However, this manuscript is not ready for publication yet. There are some issues that the authors need to address before the manuscript can be considered for publication. Detailed comments for consideration are provided below.

Comment 1#

Line 20: authors write “Notably, no difference was observed in DNA electrophoresis spectrum between desulfated agar and commercial agarose” this is not true when we look at Figure 3.

Comment 2#

Lines: 41, 159,165, 218, 260, 261, 262, 266, 334, 341, 342, 355 comma instead of dot.

Comment 3#

Lines: 98, 103, 107, 119, 176, 177, 178, 237, 244, 252, 253, 293, 350, 351, 364, 372 Improvement of the punctuation marks or add superscripts and subscripts.

Comment 4#

Figures and tables should be placed in the main text near to the first time they are cited. Please correct the placement of the figures and tables.

Line 239: Figures should appear in order of citation, why is figure 5 first and not figure 4. Please correct.

Comment 5#

Lines: 148, 186: Text ” This is a table” in table signature. You can see that this is the table this information is not needed. Correct the table signatures.

Author Response

(The authors gave the same response as above.)

Reviewer 3 Report

The manuscript (marinedrugs-1209508), titled "Convenient agarose preparation with hydrogen peroxide and desulfation process analysis," is, in my opinion, within the journal scope and interesting for the Marine Drugs readers. However, the authors need to explain the figures more carefully, particularly the capital and lower case letters. In addition, they should correct the table's caption and the lower case letters presented in table 2. Finally, they should revise the English language.

Author Response

(The authors gave the same response as above.)

Round 2

Reviewer 1 Report

Dear Authors,

I accepted all resposnes and in my opinion manuscript should be accepted in the present form. 

Best regards,